# Molecular xenomonitoring of diurnally subperiodic *Wuchereria bancrofti* infection in *Aedes (Downsiomyia) niveus* (Ludlow, 1903) after nine rounds of Mass Drug Administration in Nancowry Islands, Andaman and Nicobar Islands, India

Addepalli Premkumar[1], Ananganallur Nagarajan Shriram[2]*,
Kaliannagounder Krishnamoorthy[2], Swaminathan Subramanian[2], Venkatesan Vasuki[2],
Paluru Vijayachari[1], Purushothaman Jambulingam[2]

**1** ICMR-Regional Medical Research Centre, Department of Health Research, Ministry of Health & Family Welfare, Andaman & Nicobar Islands, **2** ICMR-Vector Control Research Centre, Department of Health Research, Ministry of Health & Family Welfare, GOI, Medical Complex, Indira Nagar, Puducherry

* anshriram@gmail.com, shriram.an@icmr.gov.in

## Abstract

A group of four human inhabited Nancowry Islands in Nicobar district in the Andaman and Nicobar Islands, India having a population of 7674 is the lone focus of diurnally sub-periodic *Wuchereria bancrofti* (DspWB) that is transmitted by *Aedes niveus* (Ludlow). Microfilaria (Mf) prevalence was above 1% even after nine rounds of Mass Drug Administration (MDA) with DEC and albendazole. Molecular xenomonitoring (MX) was conducted to identify appropriate vector sampling method and assess the impact. BioGents Sentinel traps, gravid traps and human baited double bed nettraps were used in three locations in each village to collect *Aedes niveus* female mosquitoes. Subsequently daytime man landing collections (MLC) were carried out in all the 25 villages in the islands. Collections were compared in terms of the number of vector mosquitoes captured per trap collection. Females of *Ae. niveus* were pooled, dried and processed for detecting filarial parasite DNA using RT-PCR assay. Vector infection rate was estimated using PoolScreen software. Only 393 female mosquitoes including 44 *Ae. niveus* (11.2%) were collected from 459 trap collections using three trapping devices. From 151 MLCs, 2170 *Ae. niveus* female mosquitoes were collected. The average prevalence of *W. bancrofti* DNA was 0.43%. Estimated upper 95% CI exceeded the provisional prevalence threshold of 0.1% in all the villages, indicating continued transmission as observed in Mf survey. MLCs could be the choice, for now, to sample *Ae. niveus* mosquitoes. The PCR assay used in MX for nocturnally periodic bancroftian filariasis could be adopted for DspWB. The vector-parasite MX, can be used to evaluate interventions in this area after further standardization of the protocol.

**Data Availability Statement:** All relevant data are within the manuscript in the form of primary tables and graph and its supporting information files.

**Funding:** This study was supported by a grant from the Indian Council of Medical Research, New Delhi, scheme No. Tribal/52/2010-ECD-II under Translational Mode. The grant was provided to Dr. A. N. Shriram. The funders had no role in study design, data collection and analysis, decision to publish or preparation of the manuscript.

**Competing interests:** The authors have no competing interests whatsoever.

## Author summary

Lymphatic filariasis (LF), caused by nematode parasite–*Wuchereria bancrofti*, is prevalent in 72 countries with about 1.39 billion people facing the risk of infection. In India LF is endemic in 256 districts. A physiological variant of the parasite, the diurnally sub-periodic Wb (DspWb) is confined to a small pocket of four remotely located isles, in Nicobar district in the Andaman and Nicobar Islands. The parasite is transmitted by a day-biting and forest dwelling mosquito, *Aedes niveus*. Even after 9 rounds of Mass Drug Administration under the National Programme for LF elimination, microfilaria prevalence was above transmission threshold level (1%), indicating continued transmission. We studied filarial infection in *Ae. niveus* using molecular xenomonitoring (MX). The vector mosquito was sampled using BioGents Sentinel, Gravid and human baited double bed net-traps. Since the traps did not yield adequate numbers, man landing collections were carried out. The prevalence filarial infection in mosquitoes assessed by molecular assay was above the provisional threshold level (<0.1%.), confirming that the transmission has not been interrupted. MX protocol can further be standardised for use as a surveillance tool in assessing the impact of MDA in this vector-parasite combination.

## Introduction

Lymphatic filariasis (LF) is endemic in 72 countries where about 1.39 billion people are at the risk of acquiring infection. *Wuchereria bancrofti* is the predominant parasite while *Brugia malayi* and *B. timori* are restricted in distribution [1]. Global Programme to Eliminate Lymphatic Filariasis (GPELF) launched in 2000 [2] has made a significant impact on infection and disease [3]. Transmission control with mass drug administration (MDA) (with an option of supplementing integrated vector management) and alleviation of sufferings of the diseased with morbidity management and disability prevention (MMDP) are the recommended strategies for achieving the goal of LF elimination. Transmission Assessment Survey (TAS) is the recommended protocol to evaluate the impact of the programme and take a decision on stopping the intervention [4]. Recently, molecular xenomonitoring (MX) has been demonstrated to be a potential surveillance tool to supplement TAS for different parasite and vector combinations [5–7].

In 2004, India, contributing around 44.3% of the global burden [1], launched the National programme to eliminate LF in 256 endemic districts in 21 States and Union Territories with about 610 million running the risk of contracting infection. Administration of MDA with diethylcarbamazine citrate (DEC) and albendazole (ALB) simultaneously is the main strategy to interrupt transmission besides recommending an integrated vector management, wherever feasible. *Wuchereria bancrofti* is prevalent in all the endemic States while *B. malayi* is restricted to six States and Union Territories in India. Nocturnally periodic *W. bancrofti* is prevalent in all the endemic districts. Diurnally sub-periodic *W. bancrofti* (DspWB), a physiological variant is confined to four (Chowra, Teressa, Kamorta and Nancowry) remotely located group of Nancowry Islands in Nicobar district in the and Nicobar Islands [8, 9]. These islands are known to be endemic for only DspWB. Ever since the incrimination of the day biting *Aedes (Downsiomyia) niveus* mosquito (earlier known as *Downsiomyia nivea*, *Ochlerotatus* (Finlaya) *niveus* and *Aedes* (*Finlaya*) *niveus*) as the vector of this form of filariasis in these islands [10], there has been significant advancement towards understanding the distribution and bionomics of vector mosquito [11–13]. The vector mosquito prefers to breed primarily in innumerable and inaccessible tree holes spread in the forested tracts of Nancowry islands [10–12].

As in other endemic districts, the National Filariasis Elimination Programme is being implemented in the Nancowry Islands of Nicobar district since 2004. Microfilaria (Mf) survey carried out after six rounds of MDA in 2011 to assess the impact of MDA showed that the overall microfilaria (Mf) prevalence was 3.28% in four islands [14]. As the Mf rate was more than 1%, MDA was continued and nine rounds of MDA were completed by 2014. Even after 9 rounds of Mass Drug Administration, microfilaria prevalence (1.7%) was above the transmission threshold level (1%), indicating continued transmission [15].

Mf survey is operationally feasible in this area since only day blood samples are needed. However, most of the Nicobarese go into the forest daily to collect forest products like firewood and their availability for blood sampling during the daytime remains uncertain. In such situations, monitoring LF infection in the population at risk *via* mosquitoes (MX) offers a key pathway for illustrating possible transmission as it has been recommended as an instrument for observing the impact of MDA on LF transmission [5, 6, 16–19]. MX protocol has already been developed for *W. bancrofti* parasite and other vector combinations [16, 17, 19]. Although infection in *Ae. niveus* was detected earlier using microscopy [11], the scope of MX has not yet been attempted for DspWB and *Ae. niveus* vector combination. The present study was carried out to evaluate the impact of nine rounds of MDA using MX. However, collection of the vector mosquito has been a challenge though Biogent Sentinel (BGS) has proved useful for other *Aedes* vector species elsewhere [20]. Therefore, apart from BGS, we used different mosquito sampling devices for collection of the vector species in the islands to compare and identify the most productive one. The study also aimed at assessing the vector infection by molecular assay developed for *Culex quinquefasciatus—W. bancrofti* combination [19, 21], which could be used as a surveillance tool in evaluating MDA in *Ae. niveus* transmitted *W. bancrofti* infection.

## Materials and methods

### Study area

The Nancowry group comprise seven islands (Chowra, Teressa, Katchal, Kamorta, Nancowry, Trinket & Bompoka) (Fig 1). Out of these seven, Bompoka is not inhabited by humans. Post tsunami, the inhabitants of Trinket Island have been rehabilitated in Kamorta. Katchal is non-endemic for filariasis. Therefore, the study was undertaken in the four human inhabited islands that are endemic for LF (Chowra, Teressa, Kamorta, and Nancowry Islands) between May 2014 and July 2015 [Fig 2A–2D]. The total population of the four inhabited islands is 7674 [22]. The population in these islands ranges from 713 (Nancowry) to 3757 (Kamorta). There are 25 villages in the four Islands and population in these villages ranges from 60 (Kanahinot) to 1759 (Kamorta Hqs). The total area ranges from 5.85 km$^2$ (Chowra) to 131 km$^2$ (Kamorta). The islands are predominantly inhabited by the Nicobarese tribe. The total number of households in these islands is 2026 with the average family size being 4. The Asian tsunami struck the A & N islands on 26$^{th}$ December 2004 and the rehabilitation measures including construction of permanent shelters were accomplished between 2005 and 2006. Briefly, a shelter comprises a living-cum-dining room, two bedrooms and a kitchen. The flooring is cemented concrete. There is a small veranda/porch, which has the entry to the shelter. The plinth area of each shelter measures approximately 450 sq. ft. All the shelters are provided with basic sanitary facilities. These shelters are proximal to the forest (tropical evergreen) and are prone to mosquito menace from the surrounding forest. Other than domestic containers, there are no other breeding habitats within the vicinity of shelters. The tribal community frequent the forest for their livelihood and are at the risk of getting mosquito bites. The islands are accessible only through Govt.-run ferries and all the essential commodities are transported from the mainland routed through the Andaman and Nicobar Administration. All these

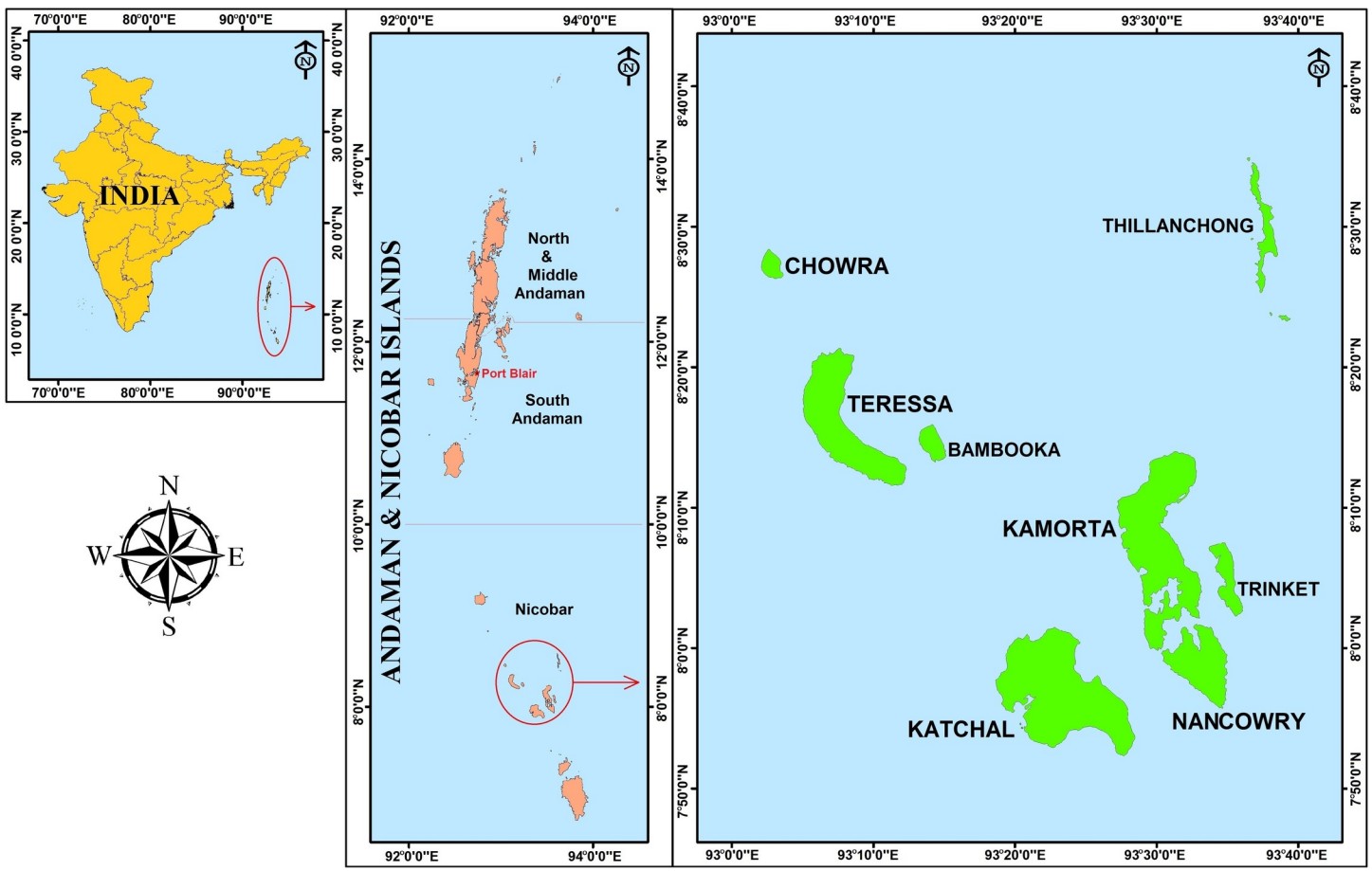

**Fig 1. Map of study area showing the location of Nancowry Islands, Andaman and Nicobar Islands, India.**

islands are within the jurisdiction of Nancowry Tehsil (sub-district), an administrative unit of a district.

## Mosquito sampling

**Collection sites.** Initially, a total of 21 villages (sites) from three islands (Teressa, Nancowry and Kamorta) were selected for mosquito sampling using BioGents Sentinel Traps (BGS, Biogents, AG, Regensburg Germany), human baited double bed net-trap (HBDNT) and gravid traps (GT). None of these devices were productive in terms of collecting adequate number of mosquitoes to reach the sample size. Subsequently, man landing collections (MLCs) were carried out in all the 25 villages in the four islands (Teressa, Nancowry Kamorta and Chowra) from December 2014 to July 2015. Three settings, the domestic (space within the human dwelling and the close surroundings), peri-domiciliary (an area under 10-meter radius from the backyard of the human dwelling) and sylvan (area within 20–25 meter radius from peri-domiciliary) settings were identified.

A permanent house is the collection spot in domestic setting (A permanent shelter/house in the study area typically had walls made up of aerated cement concrete blocks, with thick exteriors, and false ceiling under the corrugated galvanized iron (CGI) roof comprising 9mm × 4mm thick processed bamboo boards supported by steel frame, one or two bedrooms, a dining-cum-living space and a front porch/veranda. The flooring is cemented concrete. Such

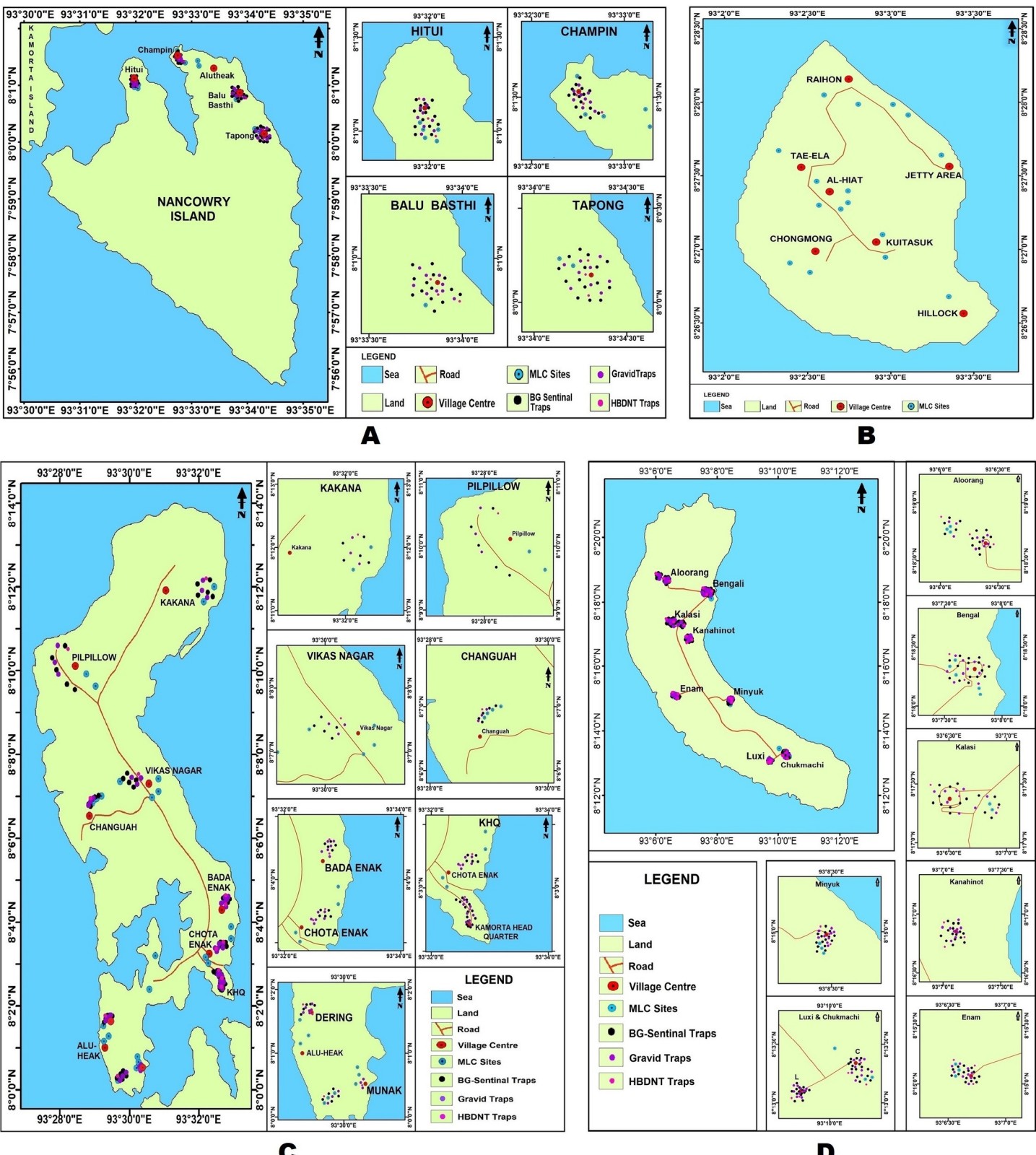

**Fig 2. Map depicting sampling locations by islands (2A: Map of Nancowry, 2B: Map of Chowra, 2C: Map of Kamorta; 2D: Map of Teressa).**

a residential structure covered by a single roof is known as a domicile. A cluster of such residential structures/permanent shelters, where the villagers live for a larger part of the day, is defined as a domestic setting).

The peri-domiciliary area is an annex of each house, consisting of a yard of about 10-meter radius. This is not connected to the domicile (porch, kitchen, bedrooms etc.), structures like community hall, *machan*-a raised platform covered on all three sides by tin sheets-structure. People spend their daytime in this area and engage themselves in processing coconut into copra. Pens for poultry like hens and chicken were constructed on wooden poles located near a tree, where chickens sleep and the chickens, which were allowed to wander freely, would nest inside the house, and underneath the foliage of the bushes located at different distances from the home constituting the peri-domiciliary setting.

A sylvan setting was defined as a location with steady forest canopy cover, contiguous to the peri-domiciliary setting. The forest canopy is constituted by tropical evergreen forests, characterized by dense forest with fruiting trees and secondary forest growth.

## Sample size

We assumed a Mf prevalence of 1%, which is the threshold recommended for conducting transmission assessment survey (TAS) for arriving at a decision on stopping MDA. Expecting random biting of mosquitoes at this prevalence level in humans, we assumed the prevalence of infection in mosquito to be at 1%. The sample size derived was 1592 mosquitoes from the four islands, with an absolute precision of 0.5% (0.5–1.5%) and a design effect of 1 at 95% confidence level.

## Mosquito sampling devices

BGS Trap (with human lure as host seeking attractant, supplied by the manufacturer) and HBDNT (with human volunteer as bait) were used for sampling host seeking (feeding phase) vector mosquitoes and GT was used to trap females attracted for oviposition (oviposition phase) with fusions as attractants. The locations and timings were fixed on the basis of the host seeking and oviposition behaviour and related activities of the vector mosquito. *Ae. niveus* is a forest dwelling, canopy habitat mosquito [23], frequenting houses in proximity to forest, zoophilic and prefers to feed on primates (primatophilic) [24]. Each of these trapping devices was placed in all the three settings (domestic, peri-domiciliary and sylvan settings).

## BG-Sentinel trap (BGS)

Battery operated BGS traps using BG sentinel lure as attractant was deployed for sampling vector mosquitoes between May and October 2014. In the domestic setting, the BGS traps were placed in the porch of the household. In the peri-domiciliary and sylvan settings, the BGS traps were hung from a tree branch 20–30 cm above the ground, as *Ae. niveus* flies low and has a propensity to bite at the feet. Grease was smeared on the cords used for suspending the trap to prevent predation of collected mosquitoes by ants. The traps were set in the morning (7.00 AM) and removed in the evening (5.00 PM).

## Gravid trap (GT)

A modified version of the battery operated CDC GT [19] was used to sample gravid mosquitoes during May-October, 2014. In each village, three traps were set in each setting. In order to maximize the trap collections, two types of infusion were used, using cashew leaves (*Anacardium occidentale*) or cumin seeds (*Cuminum cyminum*) as they were reported to have potency

**Table 1. Details of different trapping devices, numbers and duration in three different ecological settings for sampling *Ae. niveus* in Nancowry Islands.**

| Ecotopes | Type of Traps | Teressa[1] | | Nancowry[2] | | Kamorta[2] | |
|---|---|---|---|---|---|---|---|
| | | Total No. of traps | No. of days | Total No. of traps | No. of days | Total No. of traps | No. of days |
| Domestic | BGS[#] | 40 | 8 | 20 | 4 | 5 | 1 |
| Peri-domestic | BGS[#] | 40 | 8 | 20 | 4 | 5 | 1 |
| Sylvan | BGS[#] | 40 | 8 | 20 | 4 | 45 | 9 |
| **Total** | | **120** | **24** | **60** | **12** | **55** | **11** |
| Domestic | GT | 24 | 8 | 12 | 4 | 9 | 3 |
| Peri-domestic | GT | 24 | 8 | 12 | 4 | 9 | 3 |
| Sylvan | GT | 24 | 8 | 12 | 4 | 27 | 9 |
| **Total** | | **72** | **24** | **36** | **12** | **45** | **15** |
| Domestic | HBDNT[@] | 8 | 8 | 4 | 4 | 3 | 3 |
| Peri-domestic | HBDNT[@] | 8 | 8 | 4 | 4 | 3 | 3 |
| Sylvan | HBDNT[@] | 8 | 8 | 4 | 4 | 9 | 9 |
| **Total** | | **24** | **24** | **12** | **12** | **15** | **15** |

# BGS-Lure

@ HBDNT-Human

[1] GT with Cumin seed infusion

[2] GT with Cashew leaf infusion

in attracting *Aedes* spp. [25, 26]. The GTs were set in areas which were considered safe and there was no obstruction. Batteries were recharged each day and we did not observe any instances of disruption. A total of 153 trap collections spreading over 51 days were made, spending 918 hrs (Table 1). The traps were set in the forenoon between 6.00 AM and 12.00 noon.

## Human baited double net trap (HBDNT)

The assembly of the HBDNT comprised two layers of mosquito net. The first layer comprised an inner mosquito net (2.4 m length × 1.60 m breadth × 1.80 m height), which was gracefully stitched to the second layer of a mosquito net (2.9 m length × 2.10 m breadth × 1.80 m height). During the HBDNT catches, an adult volunteer, who acted as bait laid on a strong, flexible and water-resistant tarpaulin sheet, placed inside the inner mosquito net, between 8.00 AM and 1.00 PM. Potable water was always available during this period. It was ensured that the bait had his breakfast, prior to entry into the double bed net. The bait answered a call of nature during the period of collection. The inner mosquito net was neatly tucked underneath the tarpaulin sheet to keep the mosquitoes out. Thus, the bait was fully protected from mosquito bites. The gap between the first and the second mosquito net was 25 cm and was rolled up to 50 cm above the ground to allow the attracted mosquitoes to come close to the first mosquito net. After every 10 minutes, an insect collector caught mosquitoes resting in the outer net. Then, the external mosquito net was rolled up and all the trapped mosquitoes were captured with oral aspirators by insect collector. Thus, mosquitoes were collected at 10 minutes' interval. HBDNT collections were conducted between July and December 2014.

## Man landing collections (MLCs)

Collections were carried out between December 2014 and July 2015 during the peak period of abundance of *Ae. niveus* [13]. Human volunteers to act as bait were identified in consultation with the village head. Only male volunteers in the age group of 20–30 consenting to the study

participated. Collections were conducted in two sittings, one in the morning between 4:00 and 7.00 AM and the other in the evening between 5.00 and 7.00 PM, coinciding with the peak biting activity of the vector species [11]. During the period of study, there were no reported local dengue/chikungunya or zika cases in the Islands. The volunteer was made to sit on a chair, outdoors near a human dwelling, exposing both the arms below elbow and limbs below knees. Mosquitoes that landed and attempted feeding on the exposed parts were collected using oral aspirators by a trained technical staff. Utmost care was taken by not allowing the mosquitoes to bite the human bait. The native Nicobarese spend considerable part of the day working in the forests/coconut plantations for harvesting coconuts and miscellaneous food articles. *Ae. niveus* is a sylvan mosquito, breeding in tree holes in the forest and resting outdoors [10–12]. It is learnt from the local tribal community that they are bitten by mosquitoes when they frequent the forests. Anticipating that the vector species is an opportunistic biter on the people engaged in the forest, MLCs were conducted in the sylvan settings only. The duration of collection time varied on the days of intermittent rains and depending on the continuous availability of the volunteer.

## Identification of mosquitoes and processing for PCR

Mosquitoes from types of collections were transferred into test tubes, labelled with date, place and type of collection and transported alive to the field laboratory of RMRC for further processing. In the laboratory, mosquito samples were anaesthetized with ether and a trained entomologist identified the species using stereomicroscope and standard taxonomic keys [27, 28]. Technicians trained in mosquito taxonomy processed the mosquitoes. The mosquitoes were separated species wise, according to sex and gonotrophic phase (unfed, blood fed, semi-gravid) and recorded. Mosquitoes collected by MLCs in a given location and collection day were pooled separately. Each pool of mosquitoes representing the location and date of collection was considered as a sample for extraction and assay for detecting filarial infection. Female *Ae. niveus* mosquitoes were pooled in vials each with 10 mosquitoes, dried overnight using dry bath and stored at -20˚C. The samples were then transported to Vector Control Research Centre, Puducherry (an Institute under Indian Council of Medical Research and a collaborating institute for the study) for molecular assay to detect filarial parasite DNA.

## Extraction and detection of *W. bancrofti* parasite DNA

Extraction of *W. bancrofti* parasite DNA from the pooled mosquito samples was performed following the manufacturer's instructions using "DNA extraction Solution Kit Genie" (Genie-Bangalore). Real-time polymerase chain reaction (RT- PCR) assay was carried out following the technique described earlier [29] with 12.5 μl of FastStart Essential DNA probes Master (Roche Diagnostics, Germany) along with 450 nmol/L of each primer: LDR1-5'ATTTTGAT CATCTGGGAACGTTAATA-3';LDR2-5'CGACTGTCTAATCCATTCAGAGTGA-3' and 125 nmol/L probe (6 FAM-ATCTGCCCATAGAAATAACTACGGTGGATCTG-TAMRA) in a final volume of 25μl in 96-well MicroAmp optical plates (Roche Diagnostics, Germany). One microliter of the extracted DNA was used as a template in RT- PCR along with 1 ng, 100 pg and 10 pg of purified genomic DNA samples as positive controls and water negative controls. All RT-PCR reactions were run in duplicates. Cycle of quantification (Cq) values for each sample is thus a single value reflecting the cycle number used for quantification. Thermal cycling parameters used were 50˚C for 2 min, 95˚C for 10 min followed by 40 cycles of 95˚C for 15 sec and 60˚C for 1min. Thermal cycling and data analysis were done with Light Cycler® 96 (Roche, Germany) instrument using the sequence detection system (SDS) software (Applied Biosystems). Cq values of samples ranging from 1.0–39.0 were considered positive, and

samples that failed to reach the fluorescence threshold beyond 39 were considered indeterminate and repeated to confirm the negativity or positivity of those samples following standard procedures.

### Data analysis

The density of man landing vector mosquito per hour was calculated by dividing the number of *Ae. niveus* collected by the number of hours spent. The numbers of *Ae. niveus* collected in different traps were too small to compare the trap densities by statistical analysis. Therefore, the actual numbers are presented in the results section. Kruskal-Wallis one-way analysis of variance (ANOVA) was used to compare the difference in density of *Ae. niveus* in the MLCs between the villages and islands. The heterogeneity chi-square test was used to compare the pool positivity rates among islands. P value less than 0.05 was considered as significant. All statistical analyses were carried out with STATA version 14.0. The prevalence of *W. bancrofti* DNA in *Ae. niveus* (vector infection rate) was estimated using the PoolScreen software (v. 2.02) software [30, 31] from the data generated from qPCR (quantitative PCR) assays. The PoolScreen software calculates the maximum likelihood estimate of the prevalence and its 95% confidence interval.

### Institutional Human Ethics Committee Clearance

The study protocol was approved by the Institutional Human Ethics Committee of the RMRC, Port Blair. It was assured that all the necessary precautions would be followed to collect mosquitoes before probing and biting the volunteer. Written informed consent was obtained from each of the adult male volunteers who were trained for participation to act as bait.

## Results

The types of trapping devices used, their numbers and hours used in different settings for sampling *Ae. niveus* in the Nancowry Islands is furnished in Table 1. Overall, in 21 villages, a total of 235 BGS trap collections involving a total of 2350 hours (Hrs) were completed. The total number of GTs used in all the settings and villages was 153 for 918 hours. The number of traps used varied between the settings and villages and the difference was due to the availability of suitable spots in the respective settings. A total of 51 HBDNT collections were completed, spending a total of 255 hours.

### Species composition

The numbers of female mosquito species collected from BGS, GT and HBDNT (combining three ecological settings) and from MLCs are depicted in the Fig 3. The BGS trap sampled six species of mosquitoes (n = 220). *Ae. albopictus* was relatively more in number forming 40.0% of the total collected followed by *Ae. aegypti* (32.7%). Only 24 female *Ae. niveus* were collected by BGS traps, which constituted 10.9% of all the mosquitoes collected by various methods. Other species were *Ae. edwardsi* (10.5%), *Ae. malayensis* (5.0%) and *Armigeres subalbatus* (0.9%). A total of 73 female mosquitoes belonging to seven species of mosquitoes were trapped in the GT. *Ae. albopictus* (41.0%) and *Ae. aegypti* (34.3%) were the dominant species. Only 11 female *Ae. niveus* were collected. In total, 100 female mosquitoes belonging to 6 species were sampled through HBDNT. *Ae. aegypti* (44.0%) and *Ae. albopictus* (, 34.0%) were the dominant species. Only 11 female *Ae. niveus* were collected. Only a total of 2170 female *Ae. niveus* were collected from MLCs.

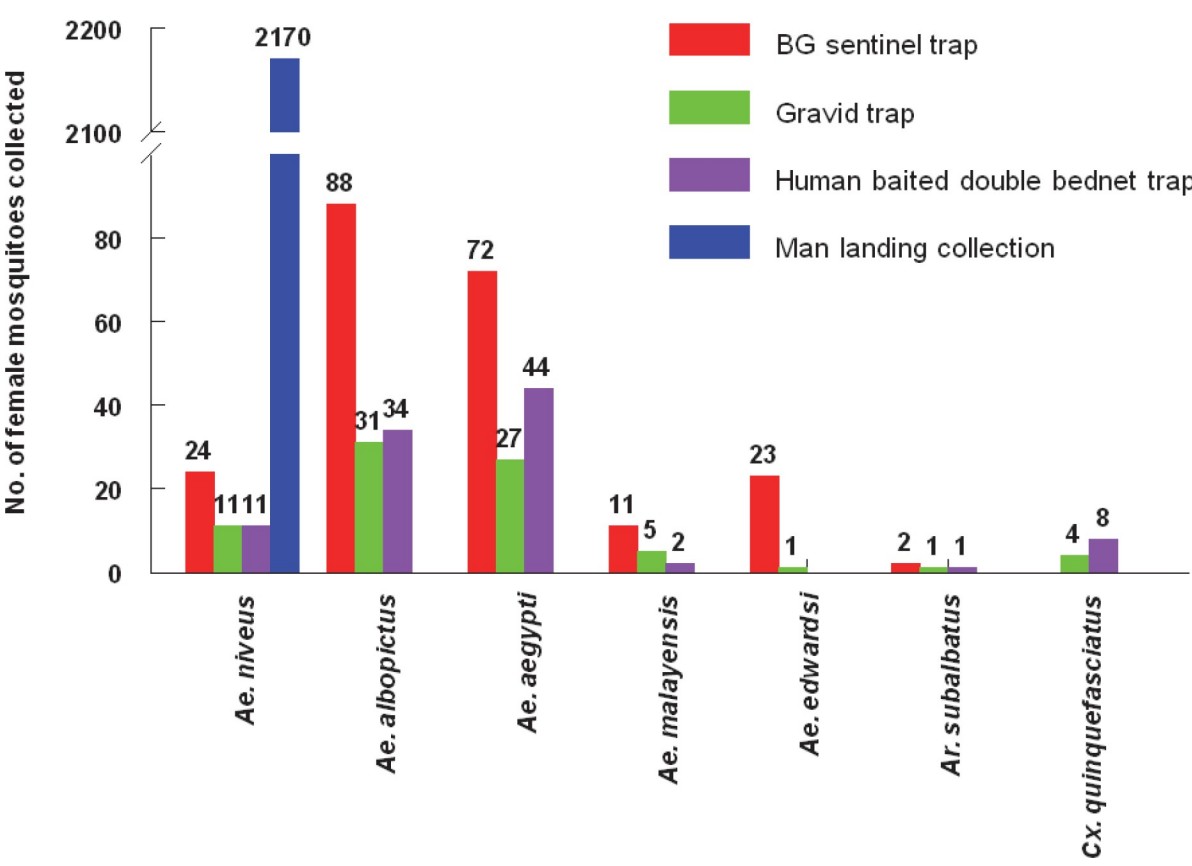

**Fig 3. The numbers of female mosquitoes captured by different traps by species.**

## MLC

As many as 151 MLCs were completed and a total of 2170 female mosquitoes of *Ae. niveus* were collected. The number of mosquitoes collected ranged from 20 to 210 in 77 sampling spots (Table 2). The density of man landing vector mosquito ranged from 1.0 to 8.0 per hour in different sites and from 1.77 to 5.68 per hour in different islands. Kruskal-Wallis one-way ANOVA showed that while the density did not differ significantly among sites ($\chi2 = 23.4$, D.F. = 24, P = 0.49), it differed significantly between islands ($\chi2 = 18.3$, D.F. = 3, P = 0.0004).

## Detection of *W. bancrofti* DNA in *Ae. niveus*

Vector mosquitoes collected from MLCs alone were processed for assessing vector infection as the number collected from trap collections were fewer. Of the 217 pools of the vector mosquito, two got damaged and 215 pools were processed by RT-PCR; 9 pools were found positive for *W. bancrofti* DNA. Pool positivity was 4.2%. Maximum pool positivity was found in Chowra (7.6%, n = 53), followed by Kamorta and Nancowry (3.70%, n = 54), while the least was in Teressa (1.9%, n = 54). The pool positivity rates were not significantly different between islands ($\chi^2 = 2.28$, P = 0.52). The pool screening calculation indicated a maximum likelihood estimate (MLE) of infection of 0.77% (95% CI: 0.25–1.86%) in Chowra, followed by 0.37% (95% CI: 0.07–1.22%) in Nancowry. The infection rate in *Ae. niveus* was the least in Teressa (0.20%, 95% CI: 0.01–0.90) (Table 3).

Vector infection was recorded only in 9 villages. The pool positivity varied between 4.8% (Raihon) and 25% (Pillpillow) in different villages sampled in the four islands. The overall

**Table 2. Number of *Ae. niveus* collected through man landing collections (MLCs).**

| Island | Villages | Period | No. MLCs | No. sampling spots | Total man hours spent | *No. Ae. niveus* collected | Man landing rate@ | Man biting rate# |
|---|---|---|---|---|---|---|---|---|
| Teressa | Bengali | April-May, 2015 | 6 | 3 | 30 | 120 | 4.00 | 48.00 |
| Teressa | Aloorang | April-May, 2015 | 3 | 3 | 15 | 60 | 4.00 | 48.00 |
| Teressa | Kalasi | April-May, 2015 | 7 | 6 | 35 | 130 | 3.71 | 44.57 |
| Teressa | Enam | April-May, 2015 | 3 | 3 | 15 | 50 | 3.33 | 40.00 |
| Teressa | Minyuk | April, 2015 | 3 | 3 | 15 | 70 | 4.67 | 56.00 |
| Teressa | Luxi | April,2015 | 2 | 1 | 10 | 50 | 5.00 | 60.00 |
| Teressa | Chukmachi | April, 2015 | 2 | 2 | 10 | 60 | 6.00 | 72.00 |
| **SUB TOTAL** | | | **26** | **19** | **130** | **540** | **4.15** | **49.85** |
| Nancowry | Champin | Dec 2014, Jan-Feb, 2015 | 12 | 3 | 60 | 140 | 2.33 | 28.00 |
| Nancowry | Balu Basthi | Dec 2014, Jan-Feb, 2015 | 17 | 4 | 85 | 170 | 2.00 | 24.00 |
| Nancowry | Tapong | Jan-Feb, 2015 | 7 | 2 | 35 | 150 | 4.29 | 51.43 |
| Nancowry | Hitui | January, 2015 | 9 | 4 | 45 | 90 | 2.00 | 24.00 |
| **SUB TOTAL** | | | **45** | **13** | **225** | **550** | **2.44** | **29.33** |
| Kamorta | Head Quarters | Dec 2014, Jan-Feb, 2015 | 12 | 4 | 60 | 120 | 2.00 | 24.00 |
| Kamorta | Chota Enak | Dec 2014, Jan-Feb, 2015 | 6 | 2 | 30 | 40 | 1.33 | 16.00 |
| Kamorta | Bada Enak | December, 2014 | 3 | 2 | 15 | 30 | 2.00 | 24.00 |
| Kamorta | Vikas Nagar | Feb-Mar, 2015 | 7 | 4 | 35 | 60 | 1.71 | 20.57 |
| Kamorta | Dering | Feb-Mar, 2015 | 7 | 4 | 35 | 70 | 2.00 | 24.00 |
| Kamorta | Kakana | Feb-Mar, 2015 | 4 | 2 | 20 | 20 | 1.00 | 12.00 |
| Kamorta | Pillpillow | Feb-Mar, 2015 | 4 | 2 | 20 | 40 | 2.00 | 24.00 |
| Kamorta | Munak | March, 2015 | 12 | 6 | 60 | 120 | 2.00 | 24.00 |
| Kamorta | Changuah | March, 2015 | 6 | 3 | 30 | 40 | 1.33 | 16.00 |
| **SUB TOTAL** | | | **61** | **29** | **305** | **540** | **1.77** | **21.25** |
| Chowra | Raihon | June-July, 2015 | 8 | 5 | 40 | 210 | 5.25 | 63.00 |
| Chowra | Kuitasuk | June-July, 2015 | 3 | 3 | 15 | 120 | 8.00 | 96.00 |
| Chowra | Tae-ela | June-July, 2015 | 3 | 3 | 15 | 50 | 3.33 | 40.00 |
| Chowra | Chongkamong | June-July, 2015 | 3 | 3 | 15 | 90 | 6.00 | 72.00 |
| Chowra | Al-hiat | June-July, 2015 | 2 | 2 | 10 | 70 | 7.00 | 84.00 |
| **SUB TOTAL** | | | **19** | **16** | **95** | **540** | **5.68** | **68.21** |
| **Overall** | | | **151** | **77** | **755** | **2170** | **2.87** | **34.49** |

@ No. of *Ae. niveus* collected ÷ total man hours spent

# No. of *Ae. niveus* collected × 12 hours

infection in the vector mosquito was 0.43% (95% CI: 0.21–0.78). Villages with no Mf carriers contain infected mosquitoes, with PoolScreen estimation, sometimes exceeding the 0.1% provisional threshold (Table 3) Also, the upper confidence limit of the vector infection exceeded the 1.0% provisional threshold in all the screened villages.

## Discussion

Although Nicobar district in the Andaman and Nicobar Islands is endemic for nocturnally periodic *Wuchereria bancrofti*, the four Nancowry Islands in the district are endemic only for DspWB with a population of 7674 at risk. Assessment by the Directorate of Health Services,

**Table 3. PoolScreen estimation of *W. bancrofti* in *Ae. niveus* mosquitoes after nine rounds of mass drug administration (DEC + albendazole) in Nancowry islands, India, 2014–15.**

| Villages | Island | Mf rate (%)$ | Mosquitoes collected | Total pools tested | No of pools positive for parasite DNA | % pools positive for parasite DNA | Prevalence of *W. bancrofti* DNA in *Ae. niveus* [95%CI]# |
|---|---|---|---|---|---|---|---|
| Bengali | TERESSA | 3.34 | 120 | 12 | 0 | 0 | 0.0 [0,2.74] |
| Aloorang | TERESSA | 0 | 60 | 6 | 1 | 16.67 | 1.66 [0.10,8.18] |
| Kalasi | TERESSA | 2.6 | 130 | 13 | 0 | 0 | 0.0 [0.0,2.56] |
| Enam | TERESSA | 0.57 | 50 | 5 | 0 | 0 | 0.0 [0.0,0.54] |
| Minyuk | TERESSA | 4.59 | 70 | 7 | 0 | 0 | 0.0 [0.0,4.28] |
| Luxi | TERESSA | 2 | 50 | 5 | 0 | 0 | 0.0 [0.0,5.540] |
| Chukmachi | TERESSA | 9.27 | 60 | 6 | 0 | 0 | 0.0 [0.0,4.83] |
| | **Sub Total** | **3.04** | **540** | **54** | **1** | **1.85** | **0.19[0.01,0.9]** |
| Champin | NANCOWRY | 0 | 140 | 14 | 0 | 0 | 0.0 [0.0, 2.40] |
| Balu Basthi | NANCOWRY | 0 | 170 | 17 | 1 | 5.88 | 0.59 [0.03,2.85] |
| Tapong | NANCOWRY | 0.69 | 140 | 14 | 0 | 0 | 0.00 [0.0,2.40] |
| Hitui | NANCOWRY | 0 | 90 | 9 | 1 | 11.11 | 1.11[0.07,5.41] |
| | **Sub Total** | **0.19** | **540** | **54** | **2** | **3.7** | **0.37[0.07,1.22]** |
| Head Quarters | KAMORTA | 0.21 | 120 | 12 | 0 | 0 | 0.0 [0.0,2.74] |
| Chota Enak | KAMORTA | 0 | 40 | 4 | 0 | 0 | 0.0 [0.0,6.51] |
| Bada Enak | KAMORTA | 0 | 30 | 6 | 0 | 0 | 0.0 [0.0,7.91] |
| Vikas Nagar | KAMORTA | 0 | 60 | 6 | 0 | 0 | 0.0 [0.0,4.83] |
| Dering | KAMORTA | 5.06 | 70 | 7 | 1 | 14.29 | 1.42[0.09,6.98] |
| Kakana | KAMORTA | 0.68 | 20 | 2 | 0 | 0 | 0.0 [0.0,10.16] |
| Pillpillow | KAMORTA | 0 | 40 | 4 | 1 | 25 | 2.47 [0.15,12.52] |
| Munak | KAMORTA | 0 | 120 | 12 | 0 | 0 | 0.0 [0.0,2.74] |
| Changuah | KAMORTA | 5.56 | 40 | 4 | 0 | 0 | 0.0 [0.0,6.51] |
| | **Sub Total** | **0.56** | **540** | **54** | **2** | **3.7** | **0.37[0.07,1.22]** |
| Raihon | CHOWRA | 2.09 | 210 | 21 | 1 | 4.76 | 0.48 [0.03,2.31] |
| Kuitasuk | CHOWRA | 5.48 | 120 | 12 | 1 | 8.33 | 0.83[0.05,4.04] |
| Tae-ela | CHOWRA | 1.96 | 50 | 5 | 1 | 20 | 1.99[0.12,9.88] |
| Chongkamong | CHOWRA | 2.91 | 80 | 8 | 1 | 12.5 | 1.25[0.07,6.09] |
| Al-hiat | CHOWRA | 3.45 | 70 | 7 | 0 | 0 | 0.0 [0.0,4.28] |
| | **Sub Total** | **2.92** | **530** | **53** | **4** | **7.55** | **0.77[0.25,1.86]** |

# Maximum likelihood estimate using PoolScreen

$ Shriram et al. 2020 microfilaraemia by village

Andaman & Nicobar administration under the *aegis* of National Vector Borne Diseases Control Programme (NVBDCP), following nine rounds of MDA indicated that the Mf prevalence was >1% in sentinel/spot check sites in Nicobar district (Personal Communication, NVBDCP data, Andaman & Nicobar Islands). Since Mf prevalence was above the pre-TAS benchmark of <1%, MDA was being continued in the district. Additionally, studies carried out in the Nancowry Islands in 2014 also showed that the microfilaria prevalence was >1% [15]. This study carried out MX as a supplementary measure to assess the impact of MDA.

Mf prevalence <1% would provide better yield for surveillance by methods like MX when it is difficult to sample human blood [5, 7, 18]. There are two components of MX, collection of vector mosquitoes and performing molecular assay. Collection of vector mosquitoes is a major challenge, particularly with species of *Anopheles* and *Aedes*. Therefore, this study assessed the efficiency of four different collection methods (three trap types and MLCs) in collecting adequate numbers to secure the sample size of *Ae. niveus* for MX. The number of *Ae. niveus*

females collected from all the three devices (BGS, GT and HBDNT) was very small. GT, which is being used for collection of *Cx. quinquefasciatus* did not seem to adapt to *Ae. niveus*, the tree-hole breeding mosquitoes. BGS-lure and HBDNT set at daytime, signalled the presence of human blood meal to the host seeking mosquitoes. Such trapping methods attracted predominantly *Ae. albopictus* and *Ae. aegypti* and very few *Ae. niveus*. Thus, the category of lures did not adapt to *Ae. niveus*. The HBDNT used human bait for attraction of mosquitoes but the bait is secured from landing and biting. Two nets as a physical barrier could have limited the entry of mosquitoes by reducing the human signal from the human bait inside the inner net and diverting to other available hosts in the proximity.

In MLCs, the mosquitoes aggressively pursue the host. Therefore, the yields are more than the HBDNT and other passive collections. The MLCs yielded collections from all the sites and in a total of 257 hours of collections, it was possible to achieve more than the minimum sample size required. However, comparison of results of MLC with those of other collections hasthe limitation that collections were conducted at different time periods and using different collecting methods. In Samoa, studies of the sampling of *Ae. polynesiensis* (diurnal) and *Ae. samoanus* (nocturnal) vector mosquitoes showed that BGS traps with any category of lure captured a greater number of mosquitoes in comparison to both CDC traps and the MLC [32]. Our study showed that MLC was better than the trap collections. This method, however, has operational and ethical issues and can be used until more productive devise is available as an alternative method.

The RT-PCR assay developed for periodic *W. bancrofti* could detect the DNA of DspWB. Lack of parallel data on vector infection assessed by dissection and microscopy limits us from commenting on the false positives. The method used in this study for DNA extraction is cheaper than the other extraction methods [21] and hence should be technically and operationally feasible. Our earlier study of vector infection based on dissection and microscopy prior to LF elimination programme in the Nancowry group of islands showed 2.65% and 0.5% infection and infectivity respectively in *Ae. niveus* mosquitoes [11]. No data on vector infection was however available during or following MDA. The present MX study showed relatively low vector infection level (0.43%) after nine rounds of MDA but still higher than the provisional threshold of 0.1% derived for *Aedes* transmitted filariasis [16, 19]. The assay detected vector infection in nine villages from all the four islands and in all these villages was more than 0.1%. The Mf prevalence in the islands ranged between 0% and 9.3% in different villages with an overall Mf prevalence of 1.7% after nine rounds of MDA [15]. In Samoa, MX conducted after seven rounds of MDA showed 4.7% vector infection in *Ae. polynesiensis* when Mf prevalence in human was as low as 0.6% [32].

In the mainland, *Cx. quinquefasciatus* is the vector of widely prevalent nocturnally periodic *Wuchereria bancrofti* and three species of *Mansonia* are the vectors of *Brugian filariasis* confined to certain pockets. In these islands, *Cx. quinquefasciatus* was the other vector species collected. But we did not consider this mosquito as it is a nocturnal biting mosquito and hence not appropriate for testing filarial infection. *Cx. quinquefasciatus*, the omnipresent and ubiquitous vector of nocturnal periodic form of *Wuchereria bancrofti* was less abundant (1.7% of the total mosquitoes collected) in our earlier study in the sites selected for the study. None of the *Cx. quinquefasciatus* dissected for filarial infection was found positive [11]. In the present study we could collect only 12 female mosquitoes of *Cx. quinquefasciatus* from gravid traps and HBDNT collections and hence were not included for molecular assay.

MX has been observed to be an indicator of human filarial infection transmitted by different mosquito vectors in wide-ranging ecological situations in American Samoa [7, 20], French Polynesia [33], Egypt [16, 34], Sri Lanka [6], Sierra Leone [35], Ghana [36] and India [5]. In American Samoa, MX results showed evidence of ongoing transmission of *W. bancrofti* by *Ae. polynesiensis* after two successful TASs, which was confirmed by TAS3 [37].

In view of the ongoing transmission, MDA was continued in Nicobar district including the four Islands with additional inputs as per the national accelerated plan of LF elimination [38]. A pilot study with mass distribution of DEC fortified salt as a supplementary measure to the ongoing MDA was implemented in two islands in 2016 and after one year of intervention total interruption of transmission was achieved [15]. MX was not carried out after DEC salt distribution and the impact was assessed only from infection in human. DEC salt intervention is now being implemented in the remaining two islands. In this context, MX can be a useful supplementary surveillance tool to TAS for evaluation of the impact as well as post-MDA monitoring. Exploring an alternative method of mosquito sampling to MLC will make MX operationally more feasible.

## Conclusions

MLC is productive in sampling day biting and forest dwelling *Ae. niveus* transmitting DspWB and other trapping devices were not efficient. However, there is a need to identify an alternative vector sampling method and standardised protocol in view of ethical concerns about involving human bait in the field. RT-PCR assay developed for nocturnally periodic WB can be used for detecting DspWB DNA. MX in *Ae. niveus* can be a supplementary to TAS and post-MDA period for early detection of risk of transmission.

## Acknowledgments

We express our sincere thanks to Dr S. K. Paul, Former Director of Health Services, Andaman and Nicobar Administration for his encouragement and support. We sincerely thank Dr H. M. Siddharaju, Chief Medical Officer, Community Health Centre, Kamorta, Dr Solomon Mark, Medical Officer, Primary Health Centre, Teressa, for extending their cooperation and assistance. We gratefully acknowledge the support extended by the Chief Tribal Council, Nancowry and Teressa Islands and the entire village Chieftains. The authors acknowledge the technical assistance rendered in the field by Mr Ajay Kumar Das, Sociologist, Mr Deepak, Mr. Shekhar Roy, Mr Barun Bacher and Mr Prabhu, Field Assistants.

## Author Contributions

**Conceptualization:** Ananganallur Nagarajan Shriram, Kaliannagounder Krishnamoorthy, Paluru Vijayachari, Purushothaman Jambulingam.

**Data curation:** Addepalli Premkumar, Ananganallur Nagarajan Shriram, Kaliannagounder Krishnamoorthy.

**Formal analysis:** Ananganallur Nagarajan Shriram, Swaminathan Subramanian.

**Funding acquisition:** Ananganallur Nagarajan Shriram.

**Investigation:** Addepalli Premkumar, Ananganallur Nagarajan Shriram, Venkatesan Vasuki.

**Methodology:** Ananganallur Nagarajan Shriram, Kaliannagounder Krishnamoorthy, Paluru Vijayachari, Purushothaman Jambulingam.

**Supervision:** Ananganallur Nagarajan Shriram.

**Writing – original draft:** Addepalli Premkumar, Ananganallur Nagarajan Shriram, Kaliannagounder Krishnamoorthy, Swaminathan Subramanian, Venkatesan Vasuki, Paluru Vijayachari, Purushothaman Jambulingam.

**Writing – review & editing:** Addepalli Premkumar, Anangananllur Nagarajan Shriram, Kalian-nagounder Krishnamoorthy, Swaminathan Subramanian, Venkatesan Vasuki, Paluru Vijayachari, Purushothaman Jambulingam.

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
