## [Decision Letter · Decision Letter 0]

30 Sep 2019

Dear Dr. SHRIRAM:

Thank you very much for submitting your manuscript "Molecular xenomonitoring of diurnally subperiodic Wuchereria bancrofti infection in Downsiomyia nivea (L) after nine rounds of Mass Drug Administration in Nancowry islands, Andaman and Nicobar Islands, India" (#PNTD-D-19-01106) for review by PLOS Neglected Tropical Diseases. Your manuscript was fully evaluated at the editorial level and by independent peer reviewers. The reviewers appreciated the attention to an important problem, but raised some substantial concerns about the manuscript as it currently stands. These issues must be addressed before we would be willing to consider a revised version of your study. We cannot, of course, promise publication at that time.

We therefore ask you to modify the manuscript according to the review recommendations before we can consider your manuscript for acceptance. Your revisions should address the specific points made by each reviewer. 

When you are ready to resubmit, please be prepared to upload the following:

(1) A letter containing a detailed list of your responses to the review comments and a description of the changes you have made in the manuscript.

(2) Two versions of the manuscript: one with either highlights or tracked changes denoting where the text has been changed (uploaded as a "Revised Article with Changes Highlighted" file); the other a clean version (uploaded as the article file).

(3) If available, a striking still image (a new image if one is available or an existing one from within your manuscript). If your manuscript is accepted for publication, this image may be featured on our website. Images should ideally be high resolution, eye-catching, single panel images; where one is available, please use 'add file' at the time of resubmission and select 'striking image' as the file type. 

Please provide a short caption, including credits, uploaded as a separate "Other" file. If your image is from someone other than yourself, please ensure that the artist has read and agreed to the terms and conditions of the Creative Commons Attribution License at http://journals.plos.org/plosntds/s/content-license (NOTE: we cannot publish copyrighted images). 

(4) If applicable, we encourage you to add a list of accession numbers/ID numbers for genes and proteins mentioned in the text (these should be listed as a paragraph at the end of the manuscript). You can supply accession numbers for any database, so long as the database is publicly accessible and stable. Examples include LocusLink and SwissProt.

(5) To enhance the reproducibility of your results, we recommend that you deposit your laboratory protocols in protocols.io, where a protocol can be assigned its own identifier (DOI) such that it can be cited independently in the future. For instructions see http://journals.plos.org/plosntds/s/submission-guidelines#loc-methods

While revising your submission, please upload your figure files to the Preflight Analysis and Conversion Engine (PACE) digital diagnostic tool, https://pacev2.apexcovantage.com/ PACE helps ensure that figures meet PLOS requirements. To use PACE, you must first register as a user. Then, login and navigate to the UPLOAD tab, where you will find detailed instructions on how to use the tool. If you encounter any issues or have any questions when using PACE, please email us at figures@plos.org.

We hope to receive your revised manuscript by Nov 29 2019 11:59PM. If you anticipate any delay in its return, we ask that you let us know the expected resubmission date by replying to this email.

To submit a revision, go to https://www.editorialmanager.com/pntd/ and log in as an Author. You will see a menu item call Submission Needing Revision. You will find your submission record there. 

Sincerely,

Audrey Lenhart

Associate Editor

Michael French

Deputy Editor

In addition to the addressing the comments from the reviewers, please note that Downsiomyia nivea is no longer the correct species name. The authors are referred to Wilkerson et al. (2015) and are requested to use Aedes niveus.

Reviewer's Responses to Questions

**Key Review Criteria Required for Acceptance?**

**Methods**

-Are the objectives of the study clearly articulated with a clear testable hypothesis stated?

-Is the study design appropriate to address the stated objectives?

-Is the population clearly described and appropriate for the hypothesis being tested?

-Is the sample size sufficient to ensure adequate power to address the hypothesis being tested?

-Were correct statistical analysis used to support conclusions?

-Are there concerns about ethical or regulatory requirements being met?

Reviewer #1: Objectives need to be more clearly defined

Reviewer #2: Objectives of the study were obscured when the authors failed to clearly visualize the context. Are they evaluating molecular xenomonitoring for infection status or intervention or both? The type of intervention they mentioned here is not discussed anywhere in the manuscript. There a number of trapping method analysis for the specific mosquito species which could also come in to the objective if they say something in the background. The sample size is not satisfactory because the prevalence they assumed is not correct. Also some analysis are not supported by the sample size. Not proper tools are followed for data analysis as these are count and possibly over dispersed. Proper definitions are necessary for the densities and rates. Proper management of volunteer engagement is not mentioned could be an ethical issue.

Reviewer #3: -Are the objectives of the study clearly articulated with a clear testable hypothesis stated?

→ Issues are raised with the stated objectives in the Peer Review comments document; authors may want to consider adding (acknowledging) what is believed to be a secondary objective

-Is the study design appropriate to address the stated objectives?

→ Study design is not clearly stated but readers can understand what the authors did, and for the stated purposes their design seems fine but more details are definitely needed

-Is the population clearly described and appropriate for the hypothesis being tested?

→ Population is described but could be more clear (though from tables it is possible to gain additional details)

-Is the sample size sufficient to ensure adequate power to address the hypothesis being tested?

→ Unclear, especially as all assumptions are not detailed; suggest this paper go to statistical review once it is in better shape (including because it is not clear if all necessary adjustments are made or if authors could have done different calculations based on their existing data – hopefully next version will clarify this to the point this can be better assessed). 

-Were correct statistical analysis used to support conclusions?

→Unclear, see above (eg they need to add more details)

-Are there concerns about ethical or regulatory requirements being met?

→Possibly – though as this was funded and performed by Indian government agencies it could be tempting to assume ethical clearance was fine. Still, authors need to add more details, including the actual clearance numbers, and especially because they used human landing catches and human baited double trap nets – neither of which are sufficiently described in detail (including the steps taken to ensure each were done with full ethical consideration and to minimize human risk/harm).

Additional Comments on Methods (from Reviewer Document to be Uploaded):

METHODS

Overall: This section needs better organization, including that some information that applies to all collected mosquitoes (e.g., processing, pooling) is currently described in one sub-section (e.g., gravid traps) and that some information in this section actually belongs in other sections (e.g., Results). ALSO please review PLoS NTD guidance on section headings (line 116 does not conform).

Study area:

1) What were the dates of this study?

2) Line 120: Here it is clear the 4 islands in the study are endemic for only DspWB, but would be helpful to state this earlier (e.g., lines 82-90, above)

3) Line 124: Please spell out or denote abbreviations or acronyms before their use (e.g., line 28)

4) What is ‘Nancowry Tehsil’? If not an important detail, can be removed (or should be defined)

5) 125-8 and 134-135: a) These seem to present similar information; suggest presenting in one section only; b) what is ‘post-tsunami’ (please state year); c) are there any other important features (e.g., standard numbers of residents, number of houses per shelter, sanitation features) that may relate to mosquito parameters?

6) Territorial area and number of houses are provided; would be helpful to have total population or perhaps better yet population density.

Sites for mosquito sampling:

1) Line 131: It appears ‘i.e.’ is used incorrectly

2) Line 132: BGS not defined yet

3) See q5, above

Mosquito sampling:

1) Some info on assumption of 1% mf prevalence (when last survey showed above 3%) would be helpful

2) Does author have a source for sample size calculation?

3) Line 140: three of ‘five’ islands or ‘four’ per elsewhere?

4) Lines 148-9: Were traps rotated? How often were nets etc. collected?

5) Line 157-63: Can authors explain when (hours set/collected), for how long (how many days), and locations (more specifics on placement)?

6) Line 172: Were gravid traps battery operated?

7) Line 174-5: How did authors check for differences in mosquito abundances as collected by the 2 different infusions (just says ‘were compared’)?

8) Lines 176-8: Can authors give more detail on where traps were set, and what does ‘closer’ mean (closer than what? Than the location of porches used for BGS?)

9) Authors state gravid traps left for 3 days; were BGS traps also left for 3 days, and if so authors should state (in any case authors should provide congruent details on each trap method) 

10) Line 183: Can authors explain why gravid traps were set between 6-12 whereas HBDNT were conducted 8am – 1pm (slight difference), could have affected results if adjusted (authors should provide timings re: BGS traps as well)

11) Lines 184-7: Several things: a) Do these methods apply for mosquito processing for all trapping method (i.e., should they be in a separate section, and not only under ‘gravid traps’ as they currently are? b) Can authors provide more detail on how species/sex were identified (e.g., by a taxonomist? entomologist? trained field staff?) [Lines 204-5 mentions stereomicroscope and taxonomic keys – is that the same for this section?] Authors could organize these sections better 

12) Line 194: Who was the ‘native Nicobarese’- was it only one person, or several people, and who was the person(s)? It is assumed a child was not used so can authors state more about the ‘bait’?

13) Line 196-7: What does this mean: ‘…attended to his individual needs…’? (eating? urinating? what? And could any of this ‘attending to needs’ have affected mosquito attraction?

14) Can authors please explain the choices of collections / trap days / hours: Lines 165 (BGS): 235 collections / 47 trap days / 2350 hours; Line 178 (Gravid): 154 collections / 51 days / 918 hours; Line 206-7: 51 HBDNT collections / 255 hours. 

15) If some traps were left several days (e.g. gravid traps for 3 days, maybe BGS if authors clarify) what methods did authors take to avoid dessication? (and, in Results, what proportion of mosquitoes was unable to be processed via PCR?)

16) Table 1: Please edit; inconsistent capitalization; abbreviations not defined (‘Gr.’) and acronyms not used (HBDNT); it’s leaF not leaVE; ‘Sylvan’ used in text ‘Sylvatic’ used in table, ‘Hours’ v ‘Hrs’ etc. etc. – this needs much cleaning up! It is authors responsibility to clean these tables, PLoS NTDs or most other journals will not copy edit tables

17) Line 210-11: These are results, should not be here; MLC info should be new (its own) paragraph –in fact lines 210-25 need seriously editing/organization to remove off-topic issues and keep on track with MLC info. 

18) Line 212-15: Is this the study period for all 4 collection types, or only MLCs? Best to clearly state when the studies took place, and state this early on in the Methods section.

19) Line 220-21: What is ‘necessary care’ – authors could include information they used to convince ethics committee; as written this is very thin on details to convey that MLC was done with appropriate consideration/caution.

20) Lines 221-25: Again, does this pertain to how all mosquitoes were processed, pooled, etc., or just the ones from MLC? There should be a section on how mosquitoes were preserved, pooled, etc. that is comprehensive or provides same information for each of the 4 types of collection methods (2 traps, HBDNT, MLC). Please see comments above re: providing more detail on how mosquitoes were handled, speciated, staged, pooled, stored, and prepared for molecular analysis especially who did these steps. 

21) Lines 229-30: Isn’t the VCRC part of the ICMR? Please sort out acronyms.

22) Line 229 and re: RT-PCR: Two things: a) Here it’s finally stated (that real-time PCR is being used); please state this in Abstract and other places requested above; b) Please also consider using acronyms consistently – RT-PCR defined line 229 then acronym is dropped by line 232.

23) Lines 253-55: What kind of ANOVA – can authors be specific? How did authors account for potential effects of using different infusions for gravid traps (and any other parameters that were different within and between methods – e.g. time of day, I if more than one ‘bait’ was used, etc.)? 

24) Line 260: define qPCR; check all acronyms in text.

25) Lines 264-66: Some of this information should be included above in Methods, including re:MLCs and HBDNT (re: informed consent, no children). The ethics section is very limited, and normally includes a study protocol/ethical approval number – can authors provide this (should be able to if funded by and conducted in conjunction with the Indian government, as it seems).

**Results**

-Does the analysis presented match the analysis plan?

-Are the results clearly and completely presented?

-Are the figures (Tables, Images) of sufficient quality for clarity?

Reviewer #1: Analysis of the Results need reorganisation

Reviewer #2: Data analysis is not sufficient and as mentioned earlier proper tools are not used. Results are not presented clearly and not described properly. The authors have to remove one table.

Reviewer #3: -Does the analysis presented match the analysis plan?

→Analysis plan not clearly or comprehensively stated; thus it is hard to assess. More details need to be added, and Methods and Results sections should be reorganized – this may help clarify on next draft.

-Are the results clearly and completely presented?

→ Results section should be reorganized – This includes that some Results (at least one line) were presented in Methods section.

-Are the figures (Tables, Images) of sufficient quality for clarity?

→ Several issues exist with Tables, mainly copy-editing and editing. Some of the tables seem to be excessive or provide detail that is unnecessary to understanding the paper. Figures and Tables were added into the main text, not always put immediately after first mention, and then added at the end of the document again - which is NOT following the submission rules of PLoS NTDs (strange the journal accepted that and sent out to Peer Reviewers in this form).

Additional Comments on Results (from Reviewer Document to be Uploaded):

RESULTS

Editing / copy-editing required: Recommend this is edited (organization) copy-edited (readability, grammar) as clearer writing could frame the results better.

1) Line 277: Ae. edwardsi actually has comparable proportion to Do. nivea, the vector of interest – could authors comment on this or provide more information here or in Discussion?

2) Line 278: First mention of ‘Ar’ without defining (spelling out) this mosquito. 

3) Lines 278-9: What does this mean - ‘were comparable’ – could authors provide more information?

4) Table 2: Please clean tables carefully- a)What is DBNT? (text refers to this throughout as HBDNT); b) how is table organized? Mosquitoes aren’t alphabetical, or by species; please consider rearranging, even putting Do. nivea first as it’s the stated mosquito of interest. 

5) Line 291: Is that significant? If authors are not considering p>0.05 then can authors clearly state this in Methods?

6) Text presents information on 6 mosquitoes but under the heading of BGS, with the following table (Table 2) showing 6 species collected via all trapping methods save MLCs. By the next text referring to 3 mosquitoes and the following table (Table 3), only 3 mosquitoes are presented via BGS. Why do authors provide further info on these 3 mosquitoes?

7) Line 300: Figure missing number; also figure if published goes immediately after paragraph where first mentioned (not at end of page). Still/again, please note submission is not supposed to include figures inside main manuscript. 

8) Check consistency in Tables and Figures; most use acronym DBT whereas text (including above in line 303 as opposed to figure in line 309) uses HBDNT.

9) Just checking – is it correct that the authors did not collect information on physiological status (e.g., unfed, bloodfed, semi-gravid, gravid)? This information should be stated, including if the pools were comprised of a mix of all females of all physiological status or if the pools included specific types (e.g., more bloodfed).

**Conclusions**

-Are the conclusions supported by the data presented?

-Are the limitations of analysis clearly described?

-Do the authors discuss how these data can be helpful to advance our understanding of the topic under study?

-Is public health relevance addressed?

Reviewer #1: (No Response)

Reviewer #2: Conclusions are supported by the data in the manuscript and are of pubic health relevant. However, it could be written in a better way. Limitations are not discussed. This article would add another evidence of the importance of molecular xenomonitoring.

Reviewer #3: -Are the conclusions supported by the data presented?

There are significant issues with the Discussion section (per PLoS NTDs section title rules, this section is called Discussion not Conclusions). This includes the very last part the authors include, called ‘Conclusions’ There is a serious lack of detail, reflection (in relation to their own work as well as contrasted by others), and discussion of limitations. These points are further flagged in the Peer Review document to be uploaded next but many are also copied below.

-Are the limitations of analysis clearly described?

Not sufficiently –see above and below

-Do the authors discuss how these data can be helpful to advance our understanding of the topic under study?

Not enough – this paper needs a more ‘global’ view on their work, including in relation to sites with other vector/parasite combinations.

-Is public health relevance addressed?

Kind of, but to be honest this seems to be more of scientific interest (rare parasite/vector combination) than of great public health interest, though the 10,000 or so people who live in the study sites could benefit greatly from better understanding of this vector/parasite combination. 

Additional Comments on Discussion (from Reviewer Document to be Uploaded):

DISCUSSION

Editing / copy-editing required: Recommend this is edited (organization) copy-edited (readability, grammar) as clearer writing could frame the results better.

1) Line 351: What is an ‘appraisal’ – what test was used? What method? Who did this? 

2) Line 352: a) What are sentinel/spot check islands? B) What is NVBDCP?

3) Line 354-5: What does this mean – ‘mf’ and ‘day samples’? If sampling in daytime is ideal, isn’t it best to detect antigen (e.g., via ICT as in TAS, with operational (daytime) feasibility being one reason WHO recommends ICT for TAS) as opposed to mf (e.g., thick blood film, filtration, etc.), which requires a night blood draw due to nocturnal periodicity for W. bancrofti? If authors suggest DspWB can be subject to daytime blood draws wouldn’t they be missing out on the rest of transmission if in A & N they also have (nocturnal periodic) W. bancrofti? if Nicobarese are not available in the day (assuming authors meant ideally they’d test for antigen via ICT) then couldn’t authors use parasitological methods such as TBFs? The arguments here are not clearly stated

4) Line 353-57: this is a non-traditional argument for MX; many often focus on the operational infeasibility of surveying humans as MDA drives prevalence (mf, antigen) levels down; it seems here authors are arguing for MX in relation to the inability to contact some Nicobarese during the daytime (i.e. practical issues) as opposed to traditional arguments of diagnostic limitations after MDA and that sampling requires more and more humans as population infection prevalence decreases. 

5) Lines 359-362: Check acronyms – MX? XM? And naming conventions - Aedes 

6) Lines 362-64: Please clarify what this means – ‘achieve the sample size’ In the end, authors did compare collection methods and commented several times on productivity. Per earlier comments authors may choose to make this a secondary objective or comment more on the findings related to their discoveries of trap type. Also here and above authors mention 3 trapping devices but could frame this as 4 different collection methods (3 trap types and MLCs)

7) Line 367: This is the first time authors mention Do. nivea is forest dwelling, despite commenting a lot on forests above (e.g., Methods and how sampling sites were chosen); authors could mention this earlier (e.g., Introduction). 

8) Lines 368-70: Please explain this logic (describe how former sentence means latter) and edit the sentences for readability.

9) Lines 373-75: These concepts are not introduced until now and should be included in Methods.

10) Line 377: What are ‘more efforts’ – please clarify, cite.

11) Line 378: Please cite some evidence/examples of the ‘matter of fact.’

12) Lines 380-83: Why is this evidence presented (presumably as a contrast to study findings) when the authors do not discuss their own findings in relation, in particular to BGS trap productivity? 

13) Lines 372-94: Please improve organization –including because concepts are unevenly and confusingly divided between paragraphs, repeated (lines 380-83 and 390-92), and presented without sufficient contrast to or reflection on relation to authors’ own results (e.g., Samoa data).

14) Line 396: What assay was developed? Do authors mean the assay they used (also please cite) and is this the first time the assay was used for DspWB (if so, authors could state this)? 

15) Lines 396-411: Would help to frame threshold and mf prevalences on same scale or help the reader interpret these; also line 402 is first mention of ‘infectivity’ so would be

16) Line 401: What is the RMRC? Please define all acronyms

17) Lines 404-6: This seems like a strong statement – what about any issues with the PCR including false negatives? There seems to be no discussion of difficulties of diagnostics, including reliability of PCR methods (including that authors seemed to use a protocol developed for nocturnally periodic W. bancfrofti but yet use it for DspWB, even if this same protocol was used in the previous study). Authors could expand on other limitations surrounding molecular analysis, including if any existed related to the speciation, pooling, kits, or any other steps or parts of the molecular analysis.

18) Lines 417-19: This sentence makes little sense in context here – why do authors present examples of general/other MX sites, discuss their results, and then discuss American Samoa (and with no citation) without contrasting to their own results?

19) Lines 419-20: How does the earlier part of sentence support the conclusion after ‘thus’ – this is a very confusing few sentences – the first part of this sentence relates to the previous sentence, the second part seems to potentially relate to the first part but the first part of this sentence but is not the sole justification for authors to conclude the second part. Also – potentially due to verb tense –as written it seems that authors use the American Samoa study results to indicate MX can be considered supplementary (in general). Suggest complete revision of this paragraph. 

20) Lines 421-26: Please revise, including for writing, punctuation, and (suggested) selection of themes presented in the final paragraph of Discussion before ‘Conclusions’

21) Line 422: Here is the first time authors mention 11 rounds of MDA; please clarify here and elsewhere, including per earlier comments (also, see title).

22) Lines 421-26: The first parts of the paragraph present different info on MDA – including introducing more round than previously stated and unpublished data on DEC salt, both of which seem important for the final conclusion about MX being a potential tool to assess transmission but neither of which are discussed in sufficient detail to support or further explain this conclusion. Also, authors may want to consider framing this idea (as they have previously) that MX can be used as a supportive or additional tool to assess transmission but still cannot be recommended as the only tool due to a variety of issues (including, but not limited to, difficulties related to diagnostics in the mosquito – something which, again, authors may want to further discuss in this section).

23) ‘Conclusions:’ Authors have previously stated their aim was not to test different collection methods, but 50% of the sentences in their conclusions focus on this, suggest reconsider framing if this was part of the aim and if so rebalance the paper or devote more time to discussion on collection methods (including limitations of each, in Discussion section).

24) Lines 434-36: Again, authors may want to take care using the term ‘alternative’ rather than ‘supplementary’ or ‘additional’ as MX literature in general, and this paper in particular (and for this vector/parasite combination) do not present sufficient evidence to support that MX can be used as the only (i.e., ‘alternative’) tool to assess if transmission is ongoing.

**Editorial and Data Presentation Modifications?**

Reviewer #1: (No Response)

Reviewer #2: The authors need to re-analysis the sample size and data analysis specified in the comments. Proper description of density/rates are lacking.

Reviewer #3: Please see my comments above and in the document I am going to upload. The presentation of data in the tables could be improved, as well as the presentation of data (including if enough data was presented) in the Results section.

**Summary and General Comments**

Reviewer #1: (No Response)

Reviewer #2: Molecular xenomonitoring is a surveillance strategy to detect circulating filarial parasites in an endemic population which is either under intervention or post-intervention. It is an important surveillance tool when human sample collection is difficult. This manuscript provided data from an endemic area where parasite load is high even after many rounds of MDA. The parasite is diurnal in that area where most of the people are outside of their house so blood collection for diagnosis is difficult. Also they undertook many sampling strategies for the specific vector which will be an important add to the scientific literature and benefit scientists and researchers of similar interest. However, the manuscript has failed to clearly portrait its objective. Their assumption for sample collection is incorrect and lacks proper analysis. Some of the analysis lacks enough data. Description could be made in a better way as the article needed to be improved by rewording.

Reviewer #3: Additional Comments - Overall, on Abstract, Author Summary, Introduction as these were not listed above as well as Final comments (from Reviewer Document to be Uploaded):

NOTE: This manuscript should be edited for English, scientific writing, and organization, as well as copy-edited for typos and grammatical errors.

Overall Comments: The paper presents interesting information, aiming to provide evidence on a parasite-vector combination that is infrequently reported in the literature as well as in a setting (islands) where elimination may be achievable (theoretically) faster than elsewhere. 

First, the entire paper should be edited and copy edited for grammar (e.g., punctuation, standard norms in capitalization, subject/tense agreement, abbreviations/acronyms defined before first use) and readability (both scientific and standard English). Better organization – between and within sections – could help the reader understand key concepts, especially in the Methods (what authors did) and Discussion (what their findings mean), that are not necessarily clear as the paper currently stands. Some small editing points include that words like viz.’ (used 7 times) and i.e., are used incorrectly; larger issues include problems with commas and other punctuation, as well as many problems with abbreviations and acronyms (incorrect, undefined before first use, or sometimes used once without being spelled out elsewhere).

Next, authors have not submitted the manuscript per instructions for PLoS NTDs - Tables and Figures are supposed to be separate/at the end? If the journal has sent paper like this to reviewers combining all elements then please note there are several mistakes: e.g., line 309 (no Figure number, figure titles should be at the bottom as opposed to table titles on top). Please check the guidance and revise / sort out.

Some important thoughts – particularly those related to mosquitoes and MX – are not carried through completely to their logical conclusions. This paper needs much better organization to help the reader follow each thought, including with the work of these authors as well as in relation or with consideration of that of other colleagues/study sites. The title of the paper is MX for post MDA surveillance, so the authors should focus on MX and the entomology but relate their findings more clearly to what this means for MDA, and if MDA can be stopped after what is currently presented as a confusing (or conflicting) number of rounds of MDA. Still, if the authors are able to address many of the comments below (as this reviewer hopes) it may strengthen the paper to warrant a recommendation of acceptance.

FOCUSED Comments:

ABSTRACT

1) BACKGROUND (lines 33-4): What does this sentence mean – could it be reworded? Isn’t MX conducted to provide follow up (to baseline, which is assumed to have provided evidence of pre-MDA infection levels in or before 2004 when MDA commenced) data on infection prevalence in humans? That is, isn’t this entire study being done in order to assess the impact of MDA in the form of DEC-fortified salt (comparing previous baseline data with this follow-up data)?

2) METHODS: Could benefit from more info on design and molecular methods (e.g. pooling details, PCR type stated clearly). Did authors test for sign cant differences in abundance and infection between islands and within villages (and if so, how)?

3) RESULTS: (line 42): ‘productive’ is a subjective term; could authors state numbers of females captured per trap type? Also, could they indicate if there were significant differences in infection between the 4 islands, or within the 25 villages (each)? 

4) CONCLUSIONS: (lines 48-99): Based on the 2 sentences in these conclusions and evidence from the study there is no compelling link or justification for this sentence to be the main conclusion. Suggest authors rephrase this section. Also, if the stated study aim was to assess progress of MDA after 12 cycles via screening for infection in mosquitoes, shouldn’t the authors present some conclusions related to what their results mean in relation to MDA based on mosquito infection – i.e., something programmatic? As written, MDA progress itself (as evaluated by MX) isn’t addressed.

5) KEY WORDS: Are some cut off and/or do authors want to consider some more (e.g., vector species, MDA, DEC-fortified salt, etc.) to help those searching key words to find their manuscript?

6) Abstract would benefit from copy-editing (some missing punctuation, subject/verb/tense agreement, and perhaps streamlined wording). 

AUTHOR SUMMARY

1) Recommend editing – among other things, several sentences don’t make grammatical sense as written (e.g., first sentence, lines 63-64) 

2) It seems ‘viz.’ is not always used correctly, including author summary (and is overused throughout this manuscript)

3) Wuchereria bancrofti could be abbreviated in second use; conversely vector name is not fully written out in first use, nor is MDA (abbreviations should be written out in here, like in Abstract, and then again redefined in first use in main text)

4) Lines 64-8: could be presented more basically (e.g., ‘filarial infection in mosquitoes was above the acceptable threshold, indicating XXX; ‘filarial in infection in humans was also above the acceptable threshold, indicating YYY’) – ideally the authors would plainly state what infection in humans and mosquitoes meant about filarial transmission in terms that the lay person would understand, rather than stating exact numbers without sufficient interpretation.’

5) Similar to comments on Abstract, authors could draw a better line between study conclusions and why MX is supportive tool for assessing MDA impact.

INTRODUCTION

Overall a good, succinct Introduction but a few suggested modifications:

1) Please check punctuation (commas and parentheses are especially misused, including those that are extra or lacking), grammar, and phrasing.

2) Lines 74-5: Please capitalize GPELF words accordingly.

3) Lines 75-6: GPELF has 2 ‘pillars:’ 1) interrupting transmission (MDA and IVC/IVM) and 2) managing morbidity; please note ‘pillar 1’ is not just MDA. It seems especially important that a paper on MX highlights the importance of the vector (IVM/IVC for interrupting transmission as well for surveillance (MX) to detect if transmission has been interrupted).

4) Lines 79-80: Is India only using MDA for ‘pillar 1’? Please clarify per point (3) above.

5) Lines 82-5: It is unclear if B. malayi and/or nocturnally periodic W. bancrofti also exist in A&N; seems DspWB exists but at this point in paper it is not clear if this is the only parasite present. [ed: line 120 states only DspWB exists in study sites but still not clear if this is only for study site (those 4 islands) – i.e. if A & N have more than 1 parasite]

6) Lines 85-90: It is unclear if other vectors exist in Nicobar (e.g. Culex, Mansonia); per this paragraph paper focuses on Do. nivea and Ae. niveus (and per title the former). Authors could present more context – e.g., if other vector/parasite combinations then what proportion of transmission is estimated attributable to those vs. Do. Nivea / DspWB. 

7) Lines 92-113: Appears that both nocturnally periodic W. bancrofti and DspWB exist, but no mention if other vectors transmit these parasites and if so if the national program targets them as well (assuming they target vector per q3, above, not just parasite via MDA)

8) Lines 93-5: Authors could briefly explain TAS (or at least that it’s WHO mandated in certain circumstances) and why this site was not eligible (e.g., include recommended thresholds) for readers who may be unfamiliar (e.g., those focused on mosquito rather than human surveillance).

9) What were the dates of this study, and of MDA? In Abstract, Methods, and Results this is not provided; from Introduction, if MDA began in 2004 and 9 annual round were given, does this mean MDA finished in 2013 or 2014? 

10) Lines 99-100: Are some words missing? 

11) Lines 104-8: This sentence is confusing (perhaps also due to commas placement) – a) what is ‘the statistically robust method’ b) suggest authors do not use ‘this form’ of filariasis but rather DspWB if that’s what they mean. 

12) Lines 108-10: Several things – a) this sentence is confusing, including phrases such as ‘…interventions such as mass DEC-fortified salt evaluations…’ (evaluations are not interventions); b) suggest authors consider softening this phrase by replacing ‘will’ with e.g., ‘may’ or ‘could’; c) here or elsewhere suggest authors include the idea that MX alone is not sufficient to indicate if transmission has been interrupted; human surveillance is still required.

13) Lines 110-11: Authors state the aim was ‘to assess vector infection…’ but title specifically states that the parasite of interest was DspWB and the vector of interest was Do. nivea – so suggest authors restate this sentence. However based on the last sentence here, as well as the Methods and Results sections, it seems that a secondary aim was to determine the best mosquito collection method); if this interpretation is correct perhaps authors could rephrase to make the aim more comprehensive.

14) Side note/small point: Authors use ‘MX’ on pages 1, 7, 8, and 9 (all pre manuscript if manuscript technically starts with ‘Introduction’) and then not again until the Discussion. They then switch to ‘xenomonitoring’ and don’t take advantage of abbreviation again until pages 31 and 32. Similarly, authors use LF in 5 instances all before page 5, and then not again. Suggest authors consider if they want to use ‘MX’ and ‘LF’ (and perhaps other acronyms) more extensively in paper after first defining them in Introduction.

FINAL Comments:

This paper needs better organization, editing (for balance – e.g., minimizing some things or removing some details or even tables, while adding other more important details in other sections), and copy editing. In particular, the paper needs more detail in the Methods and Discussion sections. Finally, the title is about MX but the authors do not focus on MX as much as they could – including general issues (including with diagnostic limitations, findings from other settings, their methods, and their study’s limitations). They also do not identify or accept that they spend significant time discussing collection methods; it may strengthen the paper to add this as a secondary aim. Still, once this is completed, the authors will b able to provide important information for a rarely studied/reported vector/parasite combination so this work would certainly add to LF knowledge in general. Also, this work is interesting for several reasons (including the use of HBDNT and MLCs that are infrequently used these days) and the fact that the work occurs on islands, which are good indications or seemingly easier places to achieve elimination than some other land-locked sites. So, this reviewer wants to convey that - despite number and detail of the above comments - the work of these authors can contribute many useful points of knowledge to the LF community and so this reviewer thanks the authors for their submission and wishes them good luck in addressing these and other peer review comments.

PLOS authors have the option to publish the peer review history of their article (what does this mean?). If published, this will include your full peer review and any attached files.

Reviewer #1: Yes: Hoda A. Farid

Reviewer #2: No

Reviewer #3: No

---

## [Editor Report · Decision Letter 1]

17 Aug 2020

Dear Dr. SHRIRAM,

Thank you very much for submitting your manuscript "Molecular xenomonitoring of diurnally subperiodic Wuchereriabancrofti infection in Aedes (Downsiomyia) niveus (Ludlow, 1903) after nine rounds of Mass Drug Administration in Nancowry islands, Andaman and Nicobar Islands, India" for consideration at PLOS Neglected Tropical Diseases. As with all papers reviewed by the journal, your manuscript was reviewed by members of the editorial board and by several independent reviewers. The reviewers appreciated the attention to an important topic. Based on the reviews, we are likely to accept this manuscript for publication, providing that you modify the manuscript according to the review recommendations. 

Many thanks for addressing the comments from the reviewers. The incorporation of reviewer feedback has very much strengthened this manuscript. There remain a few outstanding points that should be addressed:

1. Please include a description of the timing of the MLCs per village. For example, an additional column in Table 2 to say which month(s) the MLCs occurred in each village.

2. Please explain why MLCs were only conducted in sylvan areas

3. There could have been bias in assessing how effective the traps were compared to MLCs because the traps were mostly set during different months of the year (described in the text as not corresponding to the peak times for Ae. niveus) and also at different times of the day (also described in the text as not corresponding to the peak biting times of Ae. niveus). This should be mentioned/explored in the discussion.

4. Mosquitoes were clearly pooled by location, but were they also pooled based on collection day?

5. In line 422, the authors mention that Cx. quinquefasciatus were not analysed because they are day biting. Especially in this scenario where W. bancrofti is diurnally sub-periodic, it would stand to reason that day-biting Culex could also be an important vector. The authors should mention why this is not apparently the case--has it already been studies on these islands?

6. Lines 432-433 specifically mention that 2 islands that had received DEC salt achieved total disruption of transmission. Did the MX data arising from this study corroborate this finding?

7. Figure 3 (which is currently mis-labeled as Figure 1): please adjust the y-axis to better accommodate the 2170 Ae. niveus captured by MLC; as it appears now, it is confusing to have 2 separate y-axes. To avoid loss of resolution when visualizing the other values, the authors may consider adding a 'break' on the y-axis to demonstrate that the uppermost value is not in scale with the others.

8. Please edit the full manuscript for English grammar and readability.

Sincerely,

Audrey Lenhart

Associate Editor

Michael French

Deputy Editor

Many thanks for addressing the comments from the reviewers. The incorporation of reviewer feedback has very much strengthened this manuscript. There remain a few outstanding points that should be addressed:

1. Please include a description of the timing of the MLCs per village. For example, an additional column in Table 2 to say which month(s) the MLCs occurred in each village.

2. Please explain why MLCs were only conducted in sylvan areas

3. There could have been bias in assessing how effective the traps were compared to MLCs because the traps were mostly set during different months of the year (described in the text as not corresponding to the peak times for Ae. niveus) and also at different times of the day (also described in the text as not corresponding to the peak biting times of Ae. niveus). This should be mentioned/explored in the discussion.

4. Mosquitoes were clearly pooled by location, but were they also pooled based on collection day?

5. In line 422, the authors mention that Cx. quinquefasciatus were not analysed because they are day biting. Especially in this scenario where W. bancrofti is diurnally sub-periodic, it would stand to reason that day-biting Culex could also be an important vector. The authors should mention why this is not apparently the case--has it already been studies on these islands?

6. Lines 432-433 specifically mention that 2 islands that had received DEC salt achieved total disruption of transmission. Did the MX data arising from this study corroborate this finding?

7. Figure 3 (which is currently mis-labeled as Figure 1): please adjust the y-axis to better accommodate the 2170 Ae. niveus captured by MLC; as it appears now, it is confusing to have 2 separate y-axes. To avoid loss of resolution when visualizing the other values, the authors may consider adding a 'break' on the y-axis to demonstrate that the uppermost value is not in scale with the others.

8. Please edit the full manuscript for English grammar and readability.
---

## [Editor Report · Decision Letter 2]

1 Sep 2020

Dear Dr. SHRIRAM,

We are pleased to inform you that your manuscript 'Molecular xenomonitoring of diurnally subperiodic Wuchereriabancrofti infection in Aedes (Downsiomyia) niveus (Ludlow, 1903) after nine rounds of Mass Drug Administration in Nancowry islands, Andaman and Nicobar Islands, India' has been provisionally accepted for publication in PLOS Neglected Tropical Diseases.

Best regards,

Audrey Lenhart

Associate Editor

Michael French

Deputy Editor

---

## [Editor Report · Acceptance letter]

14 Oct 2020

Dear Dr. SHRIRAM,

We are delighted to inform you that your manuscript, "Molecular xenomonitoring of diurnally subperiodic Wuchereriabancrofti infection in Aedes (Downsiomyia) niveus (Ludlow, 1903) after nine rounds of Mass Drug Administration in Nancowry islands, Andaman and Nicobar Islands, India," has been formally accepted for publication in PLOS Neglected Tropical Diseases.

Best regards,

Shaden Kamhawi

co-Editor-in-Chief

Paul Brindley

co-Editor-in-Chief
